# GC-MS Coupled with Rate-All-That-Apply (RATA) to Analyse the Volatile Flavor Substances of Yellow Wine during Fermentation

**DOI:** 10.3390/foods12101992

**Published:** 2023-05-15

**Authors:** Di Xia, Xu Tan, Li Wang, Zongjun Li, Aixiang Hou, Yan Zhu, Ling Lai, Yuanliang Wang

**Affiliations:** 1College of Food Science and Technology, Hunan Agricultural University, 1 Nongda Road, Changsha 410128, China; xd328328@163.com (D.X.); m18275551420@163.com (X.T.); 15285223720@163.com (L.W.); hnlizongjun@163.com (Z.L.); aixianghou@163.com (A.H.); 2Tianjin of Industrial Biotechnology, Chinese Academy of Sciences, 32 Xiqi Road, Tianjin 300308, China; zhuyan@tib.cas.cn; 3Hunan Province Key Laboratory of Food Science and Biotechnology, Changsha 410128, China; 13787267776@163.com

**Keywords:** yellow wine, rate-all-that-apply, volatile, odor, taste, correlation analysis

## Abstract

Yellow glutinous rice wine is a traditional Chinese beverage created by soaking, boiling, and fermenting glutinous rice. The majority of current studies on the flavor of yellow glutinous rice wine are based on instrumental analysis, with sensory analysis being overlooked. In this study, 36 volatile chemicals in the fermentation process of yellow wine were annotated by GC-MS and then an OPLS-DA model was built to screen out 13 distinctive substances (VIP > 1, *p* < 0.01). The relative odor activity value (ROAV) was calculated using the threshold values of these chemicals and 10 substances, including alcohols, esters, and aldehydes, were found as key contributors to the overall flavor of yellow wine. Following that, consumers quantified the sensory descriptors of yellow wine using rate-all-that-apply (RATA), and correspondence analysis revealed three groups of characteristic flavors and odors. Alcohols and esters were found to be key producers of flowery and fruity scents in yellow wine, according to correlation analysis. We discovered two alcohols that are rarely found in yellow wine: [R,R]-2,3-butanediol and 1-phenylethanol. The former was found to be favorably connected with wine scent and pungent odor, and its specific effect on flavor should be researched further.

## 1. Introduction

Yellow rice wine (Huangjiu), also well known as Chinese rice wine, is one of the oldest wines in China with a long history of over 2500 years, and has a large domestic market [1]. Yellow wine is rich in flavor and taste and contains many ingredients with active functions [2,3,4]. Among them, polyphenols may alleviate adriamycin-DOX-induced inflammation and mitochondrial dysfunction by modulating the intestinal microbiome and the related metabolites [5]; γ-aminobutyric acid (GABA) and functional peptides are of the potential to improve memory in mice in maze experiments [6]. Additionally, yellow wine may relieve constipation by modulating neurotransmitters and intestinal flora and has been employed as a dietary supplement and pharmaceutical ingredient [7].

The flavor of Huangjiu is the key factor that impacts consumers’ acceptance and preference. Many studies have been conducted to characterize the key aroma compounds in Huangjiu [8]. In 2008, solid phase microextraction (SPME) combined with gas chromatography-olfaction (GC-O) was employed to discover 63 aroma compounds from two commercial yellow wines from Shanghai and Zhejiang [9]. Sotolon was reported as the first aroma active component in Shaoxing and Shanghai yellow wine in 2013 [10]. A recent study showed that the volatile flavor substances in yellow wine collected from 11 provinces were clustered to three groups: (i) samples from Jiangxi, Zhejiang, Shandong, and Anhui with significant diversity and complexity; (ii) samples from Jiangsu, Fujian, Shanghai, Guangdong, and Shaanxi rich in hydrocarbons; and (iii) samples from Guangxi and Liaoning rich in alcohols and esters [11]. Furthermore, Wang et al. investigated the dynamic change of aroma compounds during the brewing process of sorghum millet and yellow wine using a combination of solvent-assisted flavor evaporation (SAFE), gas chromatography-mass spectrometry (GC-MS), and chemometrics. According to studies, the primary stage of fermentation is when acids and aldehydes are created in considerable amounts, while the major stage is when alcohols and esters take center stage, and sotolon and methional were generated in the aging stage [12]. These above investigations only used instruments to analyze the primary flavor compounds in yellow wine, but they did not include flavor evaluation via manual smelling. In fact, the sensory attributes defined by consumers provide a valuable input for product analysis [13]. Therefore, it is necessary to develop a suitable sensory evaluation method for flavor analysis of yellow wines.

A descriptive sensory profile is widely applied to describe food products’ characteristics [14]. However, conventional descriptive analysis is time consuming and labor intensive [15]. Rapid sensory profiling methods have been developed, enabling the use of untrained panelists or even consumers to obtain sensory profiles of food products [16]. Among them, check all that apply (CATA) is popular and easy to implement. It is a straightforward task, for which the participants do not require any special training. Each subject is given a set of items and taught to check the qualities that apply to each product from a list of attributes [17,18]. Consumers have been reported to provide sensory spaces for many products that are very similar to those obtained using descriptive analysis with trained assessors [19]. However, CATA does not support the direct quantification of sensory attributes [20], leaving evaluation more qualitative and subjective. The rate-all-that-apply (RATA) method, a variant of CATA, was thus developed to provide quantification of sensory attributes, where panelists are asked to rate the intensity of the terms they have ticked in this variant. Consequently, RATA yields superior accuracy and better sample discrimination than CATA [21].

RATA has been increasingly used in food evaluation, such as wine, milk powder, and black tea [21,22,23], but its application in yellow wine is limited. The majority of current studies on the flavor of yellow wine are based on instrumental analysis. The odor of important volatiles is commonly used by researchers to characterize the odor of samples. This is limited since flavor compounds interact with one another. The human body has a highly developed sense of taste and smell and, therefore, manual sniffing should not be overlooked in flavor research. In this work, we used RATA to quantify the distinctive scents and tastes of yellow wine and integrated the results of GC-MS to provide a more comprehensive analysis of yellow wine flavors.

## 2. Materials and Methods

### 2.1. Yellow Wine Sample Preparation

The yellow wine fermentation was conducted according to the process shown in Figure 1. Glutinous rice was purchased from Changsha, China. Wine malt was obtained from Magujing Brewery Company (Changsha, China). Yellow wine samples (2 L) were taken using sterile plastic containers at specific timepoints (Figure 1), sealed, and stored at −21 °C for subs.

### 2.2. Annotation of Volatile Compounds

Solid-phase microextraction (SPME; Supelco, Bellefonte, PA, USA) combined with 7000D chromatography–mass spectrometry (GC-MS; Agilent, Santa Clara, CA, USA) was used to analyze the changes of volatile flavor compounds in the yellow rice wine fermentation process. A sample of 6 mL/6 g and 2 g of NaCl were added to a 20 mL extraction flask, which was then placed on a magnetically heated stirrer, and 65-μm of PDMS/DVB fiber (Supelco, Bellefonte, PA, USA) was inserted into the flask and extracted at 60 °C and 600 r/min for 45 min. Next, the SPME installation was inserted into the GC injection port at 250 °C in a non-shunt mode for 5 min to allow complete desorption. The GC conditions were as follows: HP-5MS capillary column (30 m × 0.25 mm × 0.25 mm); carrier gas: He; flow rate: 1.0 mL/min; column oven temperature: the temperature was initially 40 °C and held for 5 min, then raised to 120 °C at a rate of 5 °C/min and held for 5 min, and finally increased to 240 °C at a rate of 10 °C/min and held for 5 min. The MS conditions were as follows: voltage 70 eV, ion source temperature 230 °C, mass scanning range *m*/*z* 30–450, and electron ionization (EI) mode. 

The mass spectra of the compounds were compared with the NIST20.L mass spectra database, and those with a matching degree of >80% were extracted and analyzed. The relative content of the annotated compounds was expressed by dividing the peak area of individual substances by the total peak area [24].

The ROAV has been proposed to explain the contribution of a volatile substance to the overall flavor of a sample. Substances with ROAV > 1 are considered to make an important contribution to the overall flavor of the sample; those with 0.1 ≤ ROAV < 1 are considered to have an effect on the flavor of the sample. The ROAV calculation method was based on the ratio of a volatile component’s odor activity value (OAV) to the greatest odor activity value (OAVmax) among the volatile compounds in yellow wine [25].

The thresholds were obtained by finding the threshold of perception of each volatile compounds in grain alcohol in the book [26]. (See Appendix A for section).

### 2.3. Consumer Test

A focus group was set up. The group was invited to participate in a basic pretrial training session in which they learned about the RATA approach, test samples (F2, F3, and F4), and temporary descriptive terminology for each sample. Interim descriptive terminology and definitions were based on earlier studies on the flavor of yellow wine [27]. Participants in the process defined and settled on the final list of words, definitions, and scale anchors for each sample (Table 1).

A total of 40 participants were recruited for this study. Adults above the age of 20, with no alcohol allergy, and the ability to consume alcohol were the required qualifications. The capacity to recognize and assess fundamental taste intensity was tested in each subject.

A total of 10 mL of each sample was served in a plastic cup (30 mL) with three-digit random codes. All samples were prepared in advance of each testing session, kept fresh in the refrigerator, and then brought to room temperature for serving and assessment. For each evaluation, the samples were served in sequential order following the Williams Latin Square design to reduce presentation order and carry-over effects. All participants took a 20 min break between sessions in consideration of sensory fatigue. All 40 participants were asked to fill out the RATA questionnaire after sniffing and tasting the samples. Each questionnaire comprised 28 attributes (5 aromatics and 23 flavors) and 5-point intensity scales [28]. To prevent primacy biases, the characteristics in the questionnaire were given in a random sequence that was different for each assessor. [21]. A value of “0” was attributed by the experimenter to terms that were not rated by the participants.

### 2.4. Statistical Analysis

The relative content of flavor compounds in yellow wine at the specified timepoints were determined using the following one-way analysis of variance (ANOVA) and SPSS Statistics 25.0 software. Correspondence analysis (CA) was used to analyze the characteristic flavor and taste of yellow wine. In order to examine straightforward two-way (and multi-way) tables with some degree of connection between the rows and columns, the usage of the CA method was proven to be a suitable descriptive or exploratory tool [29]. The relationship between volatile chemicals and sensory qualities was assessed using Pearson’s correlation coefficient. Principal component analysis (PCA) and orthogonal partial least squares discriminant analysis (OPLS-DA) were carried out using SIMCA-P 14.0 (Umetrics, Ume, Sweden). To reduce variance, the experiment was run in three duplicates. The mean standard error is used to represent all values. *p* < 0.05 was chosen as the level of statistical significance. Plotting was undertaken with OriginPro 7.5 (OriginLab, Northampton, MA, USA), Chiplot online, and Cytoscape.

## 3. Results and Discussion

### 3.1. Changes in Volatiles during the Fermentation of Yellow Wine

Yellow wine contains a variety of volatile substances, which are mainly derived from raw materials and the metabolites of microorganisms [30]. In the present study, 36 volatile substances were annotated during the fermentation of yellow wine (Table 2), and all these compounds could be classified into seven groups (Figure 2A), including 13 esters, eight alcohols, four aldehydes, one phenol, one acid, one ketone, one ether, and seven other substances (e.g., alkanes, naphthalenes, azoles, nitrogenous compounds).

It was found that aldehydes and esters are major of volatile compounds in F0 (Figure 2B). Aldehydes reduced from 36.8% to 1.77% with the progress of fermentation. This phenomenon might be caused by the conversion of aldehydes into alcohols during the fermentation process. Furthermore, some of the aldehydes combined with higher alcohols to form acetals [12]. Esters increased in F2 and F3 but decreased sharply by 22.59% after filtering (F4). Alcohol rose from F0 and maintained high relative content in F2, F3, and F4. The esters kept decreasing, especially after filtering. Previous research has indicated that alcohols are more persistent than esters during the fermentation of yellow wines, and our findings support this [31]. This is said to be because esters are less soluble in water and ethanol, hence esters are lost with the lees during pressing, resulting in a quick decrease in their relative content [12]. As a result, the pressing procedure for yellow wine should be altered to retain more of the wine’s essential esters and therefore enhance its flavor.

Acids play an important role in the flavor of most fermented foods. In this investigation, the sole acid associated with yellow wine was acetic acid, which began to be generated in F3 and rose to 24.46% in F4. It has been reported that over the duration of 3–5 days of yellow wine fermentation, lactic acid bacteria increased from 2.6 lg to 5.43 lg [32]. This indicates that acids began to accumulate in the yellow wine during this period, as a result of the proliferation of lactic acid bacteria.

The study Indicated a considerable variation in the flavor material composition of yellow glutinous rice wine and yellow sorghum wine. After 5 days of fermentation, the sorghum wine was dominated by acids and esters, with a little alcohol. By contrast, glutinous rice wine contains more esters and alcohols, which may make sorghum wine taste thinner than glutinous rice wine [33,34]. Previous research has also discovered that changing the types of grains and soaking durations changes the amino acid and organic acid content of yellow wine, which influences its flavor [35]. As a result, research into the variations in volatiles across various grains must be expanded in order to improve the categorization of yellow wines.

Figure 2C shows a clustering analysis of the volatiles, which allows for a better visual portrayal of each compound’s high and low content. To avoid the problem of substantial variances in the relative content of volatile compounds, it was normalized before charting. Volatiles can be broadly categorized into five groups throughout the fermentation process, as illustrated in the graph: (1) growing trend during fermentation, including [R,R]-2,3-butanediol, acetic acid, 1-butanol,3-methyl-,acetate, and pentane,1-(1-ethoxyethoxy)-.; (2) rising and then decreasing, including ethyl tetradecanoic, ethyl oleate, ethyl caprylate, and many other esters; (3) falling and then rising, including ethyl caprylate, [R,R]-2,3-Butanediol, and phenylethyl alcohol; (4) falling trend; (5) only evident at one stage, such as (-)-globulol, phenylacetaldehyde, and 1-octen-3-ol, which were only detected in F2.

### 3.2. Screening for Characteristic Compounds in the Fermentation of Yellow Wine

PCA was used to examine and evaluate the volatile compounds of the samples from various fermentation times [1]. The results of the score plots (Figure 3A) show that the samples in each group are close to each other, proving a good reproducibility of the results. In addition, F2 and F3 are clearly dispersed, proving that there is a clear difference in volatiles between the two groups. Expectedly, there is no discernible change between F0 and F1. F4 is, surprisingly, less distinct from these two groups. The comparison shows that F4 has a fairly homogenous material type since it reduces a lot of volatile chemicals. As a result, F4 differs from F0 and F1 less in terms of material type. The amount of alcohols and acids that F4 gathered, combined with the relative content, must have affected the flavor of the yellow wine. The distribution of volatile chemicals in the various groups is displayed by the bioplot (Figure 3B).

The differential compounds in the fermentation process were screened using the OPLS-DA in conjunction with the Variable Importance in Projection (VIP) [36]. An OPLS-DA model with good fitting parameters (R^2^X = 0.986, R^2^Y = 0.996, Q^2^ = 0.99) was used in this work to screen for signature compounds that contribute to changes in yellow wine scent during the fermentation process. VIP calculated the contribution of each volatile substance to the classification. Substances with VIP > 1 were typically thought to play an essential influence in sample classification. In the OPLS-DA model, 13 volatiles were finally screened (VIP > 1, *p* < 0.01) (Figure 3C,D), including: nonanal (H2), decanal (H3), 1-hexanol (H6), [R,R]-2,3-butanediol (H7), 1-phenylethanol (H9), phenylethyl alcohol (H10), ethyl tetradecanoic (H14), ethyl hexadecanoic (H15), ethyl acetate (H24), 2,4-di-tert-butylphenol (H27), acetic acid (H28), heneicosane (H30), and 1-chloropentane (H31). Among these, phenylethyl alcohol and acetic acid have been identified in previous studies as important aroma compounds in yellow wine [11].

### 3.3. Key Volatile Compounds in Yellow Wine Fermentation

Figure 4 shows the ROAV of the characteristic flavor compounds. Aldehydes with lower threshold values contributed more to the flavor of yellow wine throughout the fermentation process. Nonanal and decanal were the key flavor substances at each stage. Although there is only one acid in the fermentation process, its influence on the flavor of yellow wine cannot be ignored. Acetic acid is a common volatile organic acid. It is one of the main flavor substances in the F3 and F4.

Alcohols were also more abundant in yellow wine, and their content increased significantly at the early stage of fermentation. The threshold of alcohols is relatively higher than aldehyde, but it plays an important role in enhancing the overall flavor of yellow wine. The four characteristic esters screened were ROAV > 1, indicating that they are key volatile flavor substances in the yellow wine fermentation process. One of the most frequent alcohols in yellow wine is phenylethyl alcohol, which is mostly created via an amino-acid-based metabolic pathway [37]. Phenylalanine is decarboxylated to produce an aldehyde, which is then converted to alcohol. In addition, the 1-Hexanol, [R,R]-2,3-butanediol, and 1-phenylethanol found in this study have rarely been recorded, and more research is needed to evaluate their effect on the flavor of yellow wine.

Esters, as the most common volatile substances in yellow wine, were detected throughout the fermentation. In red yeast yellow wine, ethyl butyrate and ethyl laurate contribute significantly to the flavor; in sorghum yellow wine, the more abundant esters are mostly ethyl lactate and ethyl phenylacetate [38]. These esters were discovered in this study as well, although they were not key flavor compounds and contributed less to the overall flavor. The key esters we found in the pair were ethyl tetradecanoic and ethyl hexadecanoic, which contribute significantly to the flavor of yellow wine at all five stages. In addition, ethyl acetate, a common ester, has a relatively high content in F3, but does not contribute as much to flavor as other compounds due to its high threshold.

### 3.4. Comparison of Yellow Wine Descriptions by RATA

F2 in the study was mainly described by consumers using the cereal, floral, wine, honey and sweet, and sour in RATA questions (Figure 5A,B). By reducing replies to two levels, the RATA findings may be transformed to CATA to get frequency data. This is regardless of its intensity rating: zero if the characteristic was not chosen as being appropriate for describing the focal sample, and one if it was [20]. Floral and sweet terms received the highest mean scores in intensity evaluation (Figure 5C,D). Similarly, the terms pungent, pineapple, wine, honey and sweet, sour, and spicy showed the highest frequency of use in RATA questions and the pineapple and spicy obtained highest average scores in intensity of sensory attributes (F3: Figure 5). Contrastingly, the terms wine, apple, pineapple, pungent, sweet, sour, and spicy were the most relevant terms for describing F4 in the study. The highest average scores of these terms were for wine and sour. Wine, sweet, and sour were mentioned several times in the description of all three samples, and it was difficult to distinguish the differences between the samples without combining the intensity scores. The average scores showed that the intensity of wine and sweet attributes gradually increased in F2, F3, and F4, with sweet being the strongest in F2. The intensity of sweet was the strongest in F2, and the sweetness decreased significantly with the extension of fermentation time, while the sourness and bitterness gradually increased.

### 3.5. Correspondence Analysis Applied to RATA

The previous section shows that there are differences in the flavor substances of yellow wine at stages F2, F3, and F4 by PCA, which will certainly cause differences in the sensory characteristics of yellow wine. Therefore, the data from RATA were analyzed using CA analysis to visualize the different odor and taste characteristics of yellow wine at the three stages. It showed that, for odor, the most heterogeneous attributes in the group are mushroom, metal, paraffin, and glue, suggesting that these odors are not representative. Honey was the odor with the least variability between groups (Figure 6A). F2 was characterized by the odors: floral, grass, banana, and cereal. The pear, pungent, and cream odors mainly characterize the F3 through the RATA. Overall, F3 exhibits a distinctly fruity aroma. Orange, pineapple, and grape aromas can also be used to describe its characteristic scent. Similarly, the F4 was characterized by wine, lemon, fat, apple, and maltose terms. As for taste, the descriptors for three samples promote the detection of a classification pattern. The salt and sweet tastes clearly characterize F2. F3 and F4 were characterized by spicy, bitterness, and sour taste, respectively (Figure 6B).

### 3.6. Correlation Analysis between Sensory Characteristics and Volatile Flavor Compounds

Based on these findings, a correlation analysis was carried out between the characteristic flavor and taste descriptors of yellow wine and key volatile flavor substances to assess their correspondence. A network diagram was drawn to show the correspondence between them (Figure 7). There is a distinct fragrance difference between the lower fatty aldehydes and the C8-C11 aliphatic aldehydes. The former has an unpleasant green odor, whilst the latter has a citrus fruit aroma [39]. Nonanal and decanal are frequently found as potentially active odor compounds in wine by GC-MS and were also annotated in this investigation. We discovered a high (r > 0.7) association between nonanal and grassy scents, lemon fragrances, and grain aromas. Decanal, on the other hand, has a substantial positive link with the scents of orange and cream. Honey, grape, and pear scents were adversely linked with the sequence. It has been demonstrated that phenylethylaldehyde can fully reduce the fruitiness of red wine, although it is unknown whether nonanal and decanal have the same effect [40].

Ethyl ester is mainly produced by the condensation of ethanol and acyl coenzyme a [41]. We found a total of nine ethyl esters in yellow wine, including ethyl octanoic, ethyl decanoic, ethyl dodecanoic, etc. Among these, ethyl tetradecanoic, ethyl hexadecanoic, and ethyl acetate contributed more to the overall flavor of yellow wine. It was shown that ethyl esters can provide floral and fruit aromas to the wine. Ethyl hexadecanoic mainly produces berry-like aromas [42]. Ethyl tetradecanoic and ethyl tetradecanoic, in this study, showed highly aromatic odor characteristics and both showed significant positive correlations with cream, grape, pineapple, and floral aromas, and strong negative correlations with lemon and wine aromas. However, there was little association between ethyl acetate and floral scents; instead, banana and pineapple scents showed the strongest positive association.

The formation of phenylethyl alcohol in wine is facilitated by a higher temperature, which increases the floral scent of the wine [43]. Yellow wine was fermented at 28–30 °C in the current study, which may raise the relative content of phenethyl alcohol and improve the floral odor of yellow wine. Correlation analysis revealed that phenethyl alcohol was indeed positively connected with floral and fruit scents, supporting the previous view. Theoretically, the ROAV of phenethyl alcohol at F3 was substantially higher than that of F2, therefore F3 should have had a more noticeable floral aroma. However, RATA data proved otherwise. This means that ROAV cannot be used to determine the final characteristic odor of the samples but can only serve as a reference. Indeed, interactions between compounds can change their original odor. For example, interactions between esters can improve the flavor of white wines, and adding various concentrations of ethyl phenylacetate to fruit notes can disguise fruit flavors while increasing floral scents [44].

The alcohols [R,R]-2,3-butanediol and 1-phenylethanol are rarely reported in yellow wine. The 1-phenylethanol contains an aromatic ring, which is the main aromatic volatile in teff, and its formation may be related to microorganism enzymes [45]; [R,R]-2,3-butanediol was strongly positively correlated with wine aroma here, weakly positively correlated with orange and pungent odors, and negatively correlated with several fruit aromas. Future research should focus on the influence of both on the flavor of yellow wine and the mechanism of its creation.

Acetic acid has been identified as one of the primary chemicals responsible for the flavor of yellow wine. Acetic acid gives yellow wine a lovely odor at the proper concentration, but at too high a content, a strong sour flavor arises [37]. Acetic acid is found in both F3 and F4, and correlation research suggests that it and [R,R]-2,3 -butanediol are the main sources of yellow wine’s strong odor. This intense odor, however, is not objectionable and rather adds to the distinctiveness of yellow wine.

For taste, bitterness, spiciness, and acidity were strongly correlated, all three were significantly positively correlated with [R,R]-2,3-butanediol and acetic acid, and significantly negatively correlated with 1-hexanol, phenethyl alcohol, and ethyl palmitate, although sweetness was the opposite of these. Specific studies are needed to show whether volatile substances have an effect on the taste of food.

## 4. Conclusions

This study investigated the changes in volatiles during yellow wine fermentation and analyzed key flavor substances that contribute to sensory attributes. It provided insight into the flavor chemistry of yellow wine. The characteristic odors and flavors of yellow wine at different fermentation times were evaluated using the RATA, and their correlation with key flavor substances was explored for a comprehensive evaluation of yellow wine flavor. We found that the results of RATA were generally consistent with previously reported results, and thus it can be used as a complementary method for flavor studies. In addition, the interaction of flavor substances can change the perception of food odors. Therefore, we cannot ignore the importance of human co-sniffing for flavor evaluation when analyzing food flavor substances. In the future, there should be more simple and effective sensory methods applied to the study of food flavor. The synergistic effect between different aroma compounds can also be further investigated, which can characterize the flavor of yellow wine more accurately.

## Figures and Tables

**Figure 1 foods-12-01992-f001:**
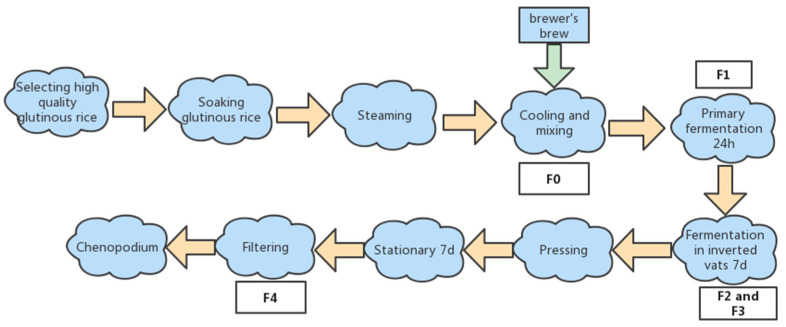
Brewing process of yellow glutinous rice wine. The sampling stages of yellow glutinous rice wine brewing were coded as follows: addition of wine malt (F0); fermentation at 28–30 °C for 24 h (primary fermentation stage, F1); after F1 is finished, the tank is inverted and fermentation continued for 3 days, F2; sampling is conducted at 5 days of fermentation, F3; fermentation until the end of the 7th day, then the wine is pressed and filtered after a week of resting, at which point the sample is taken as F4.

**Figure 2 foods-12-01992-f002:**
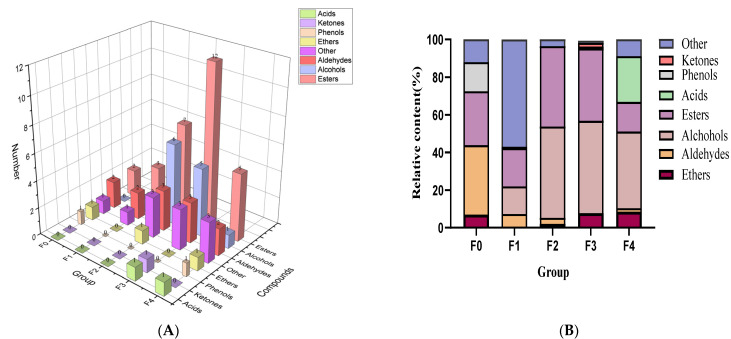
Numbers and relative content percentages of volatile compounds identified during brewing process of yellow wine: (**A**) numbers of different types of volatile compounds in different steps; (**B**) relative content of different types of volatile compounds in different stages; (**C**) clustering heat map of the relative content of 36 volatile compounds in glutinous rice Huangjiu at each brewing stage.

**Figure 3 foods-12-01992-f003:**
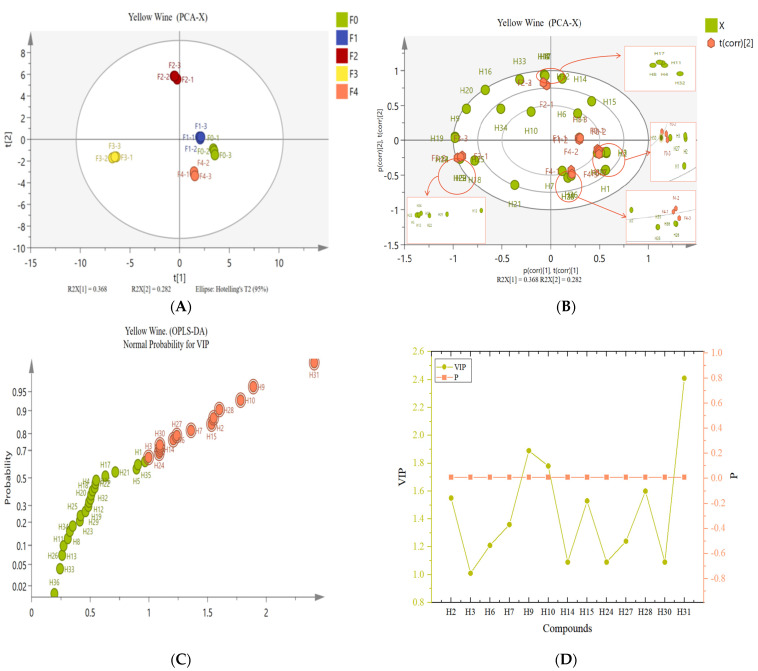
The score of PCA results at different fermentation periods: (**A**) the bioplot of PCA; (**B**) OPLS-DA diagram of 36 volatiles; (**C**) the red marked dots are the volatile compounds that have changed significantly during the fermentation process of yellow wine. The farther away from the origin, the greater the difference. The *p*-value of characteristic compounds (**D**).

**Figure 4 foods-12-01992-f004:**
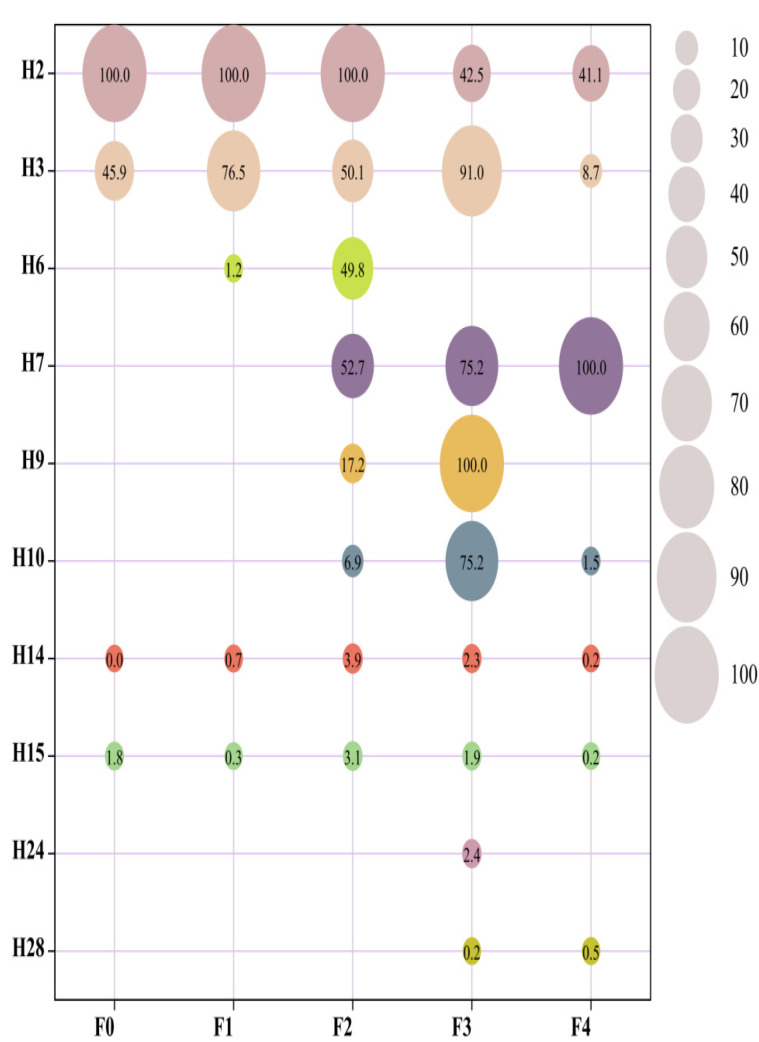
The ROAV for the 10 characteristic volatiles (H27, H30, and H31 were not queried for thresholds). The higher the ROAV, the greater its contribution to the overall flavor.

**Figure 5 foods-12-01992-f005:**
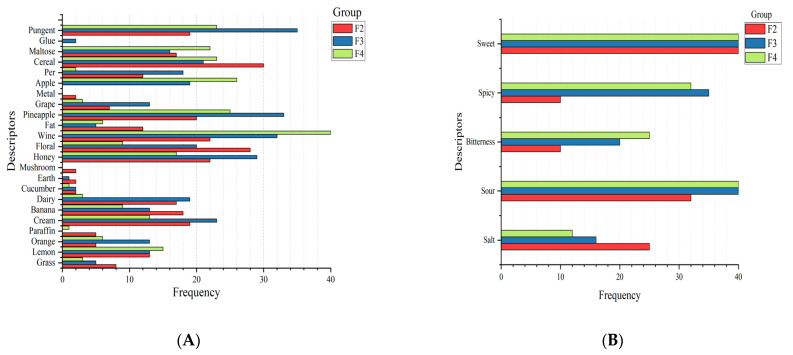
Descriptors created with RATA method grouped by odor and taste. Respective sum of frequencies from different fermentation periods in yellow wine are: (**A**) odor; (**B**) taste. The mean intensity ratings of three samples obtained for RATA method grouped by odor and taste (**C**) is odor, (**D**) is taste; a,b,c indicate significant differences between groups (*p* < 0.05).

**Figure 6 foods-12-01992-f006:**
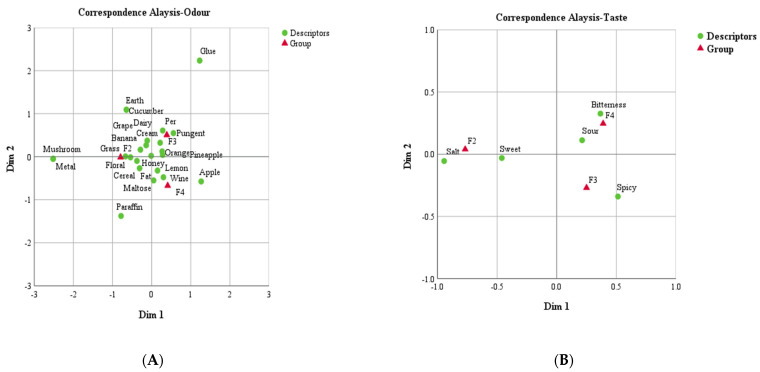
CA to compare the descriptors in each group (F2, F3, F4): (**A**) CA for odor; (**B**) CA for taste. Descriptors are represented by green dots; group is represented by red triangles.

**Figure 7 foods-12-01992-f007:**
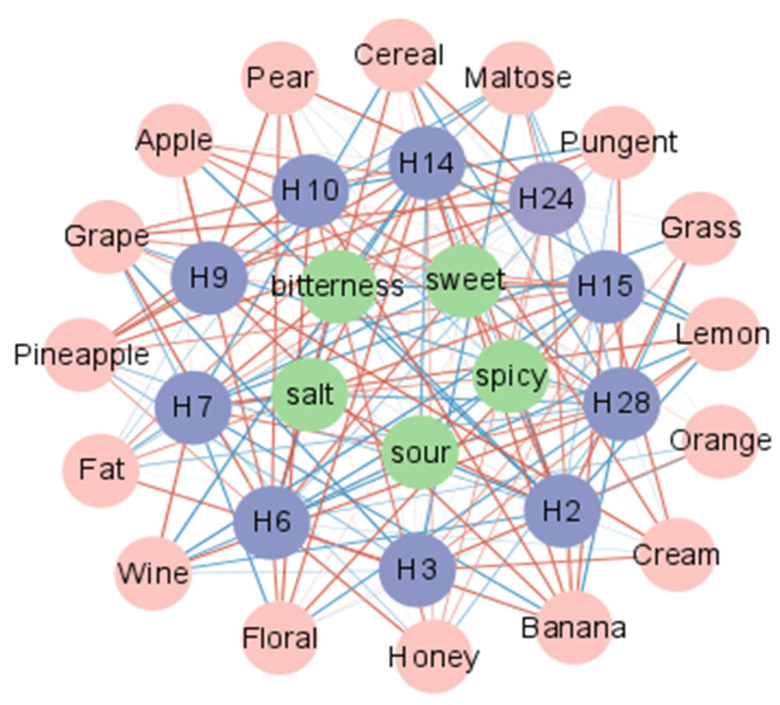
Correlation analysis network map between the sensory descriptors and the volatile compounds that change significantly during the fermentation of yellow wine. The pink circles represent odor descriptors, the green circles represent taste descriptors, and the blue circles are key flavor substances. The red line indicates a positive correlation, the blue line indicates a negative correlation, and the darker the color, the more significant the correlation.

**Table 1 foods-12-01992-t001:** Descriptors and definitions used for rate-all-that-apply (RATA) evaluations of yellow wine.

	Descriptors	Definition	Intensity				
			1	2	3	4	5
Odor	Grass	The aromatics associated with green grass					
	Lemon	The smell associated with lemon					
	Orange Peel	The smell like orange peel					
	Paraffin	The smell like paraffin wax					
	Cream	The aromatics associated with sour cream					
	Banana	The characteristic of banana vanillas					
	Dairy	The aromatics associated with the flavor of boxed milk					
	Cucumber	The fresh aroma associated with cucumbers					
	Earth	The smell associated with moist earth					
	Mushroom	The aromatics associated with fresh mushroom					
	Honey	The flavor associated with honey					
	Floral	The floral smell associated with flowers					
	Wine	The alcohol odor associated with wine					
	Fat	The aromatics associated with slightly oxidized fats and oils					
	Pineapple	The aromatics associated with pineapple					
	Grape	The characteristic of grape vanillas					
	Metal	The aromatic associated with metals, tin, or iron					
	Apple	The smell associated with apple juice					
	Pear	The aromatics associated with pear					
	Cereal	The smell of grains					
	Maltose	The sweet smell similar to maltose					
	Glue	The smell associated with glue, like formaldehyde					
	Acetate	The odor similar to acetic acid					
	Pungent odor	The odor is pungent					
Taste	Salt	The basic taste, perceived on the tongue, stimulated by sodium salt					
	Sour	The basic taste, perceived on the tongue, stimulated by acids					
	Bitterness	The astringent taste associated with tannins					
	Spicy	The basic taste, perceived on the tongue, stimulated by spicy flavors					
	Sweet	The basic taste, perceived on the tongue, stimulated by sugars					

“1 = very weak”,“2 = weak”,“3 = medium”,“4 = strong”,“5 = very strong”.

**Table 2 foods-12-01992-t002:** GC-MS detected volatiles in yellow wine during five fermentation stages.

NO.	Compound Name	Formula	CAS	Relative Content (%)				
				F0	F1	F2	F3	F4
H1	Dimethyl Ether	C_2_H_6_O	115-10-6	6.99 ± 0.66 b		2.00 ± 0.05 c		8.53 ± 0.71 a
H2	Nonanal	C_9_H_18_O	124-19-6	25.79 ± 2.20 a	4.2 ± 0.10 b	0.94 ± 0.07 c	0.20 ± 0.01 c	1.48 ± 0.09 bc
H3	Decanal	C_10_H_20_O	112-31-2	11.05 ± 1.25 a	3.0 ± 0.12 b	0.44 ± 0.04 c	0.40 ± 0.01 c	0.29 ± 0.03 c
H4	Phenylacetaldehyde	C_8_H_8_O	122-78-1	N.D	N.D	1.88 ± 0.07 a	N.D	N.D
H5	Ethane, 1,1-diethoxy-	C_6_H_14_O_2_	105-57-7	N.D	N.D	N.D	7.17 ± 0.04 a	N.D
H6	1-Hexanol	C_6_H_14_O	111-27-3	N.D	14 ± 0.80 a	5.39 ± 0.26 b	N.D	N.D
H7	[R,R]-2,3-Butanediol	C_4_H_10_O_2_	24,347-58-8	N.D	N.D	2.59 ± 0.18 b	1.85 ± 0.10 c	18.72 ± 0.33 a
H8	1-Octen-3-ol	C_8_H_16_O	3391-86-4	N.D	N.D	0.74 ± 0.04 a	N.D	N.D
H9	1-Phenylethanol	C_8_H_10_O	98-85-1	N.D	N.D	12.23 ± 0.36 b	35.7 ± 0.44 a	N.D
H10	Phenylethyl Alcohol	C_8_H_10_O	60-12-8	N.D	N.D	26.84 ± 0.42 a	9.06 ± 0.19 c	21.98 ± 1.30 b
H11	(-)-Globulol	C_15_H_26_O	489-41-8	N.D	N.D	0.65 ± 0.00 a	N.D	N.D
H12	2,3-Butanediol	C_4_H_10_O_2_	513-85-9	N.D	N.D	N.D	2.26 ± 0.12 a	N.D
H13	Ledol	C_15_H_26_O	577-27-5	N.D	N.D	N.D	0.66 ± 0.66 a	N.D
H14	Ethyl Tetradecanoic	C_16_H_32_O_2_	124-06-1	3.45 ± 0.53 c	9.4 ± 0.27 b	11.51 ± 0.14 a	3.40 ± 0.13 c	2.32 ± 0.06 d
H15	Ethyl Hexadecanoic	C_18_H_36_O_2_	628-97-7	25.11 ± 4.40 a	10. ± 0.31 b	22.45 ± 0.21 a	6.76 ± 0.24 b	6.19 ± 0.54 b
H16	Ethyl Oleate	C_20_H_38_O_2_	111-62-6	N.D	0.4 ± 0.13 c	2.52 ± 0.22 a	1.84 ± 0.00 b	N.D
H17	Hexyl Formate	C_7_H_14_O_2_	629-33-4	N.D	N.D	2.73 ± 0.02 a	N.D	N.D
H18	Ethyl Octanoic	C_10_H_20_O_2_	106-32-1	N.D	N.D	1.08 ± 0.06 c	2.65 ± 0.06 a	2.09 ± 0.02 b
H19	Ethyl Decanoic	C_12_H_24_O_2_	110-38-3	N.D	N.D	0.73 ± 0.05 b	2.32 ± 0.05 a	N.D
H20	Ethyl Dodecanoic	C_14_H_28_O_2_	106-33-2	N.D	N.D	1.67 ± 0.07 b	1.99 ± 0.11 a	N.D
H21	1-Butanol, 3-methyl-, Acetate	C_7_H_14_O_2_	123-92-2	N.D	N.D	N.D	2.98 ± 0.02 b	4.56 ± 0.73 a
H22	Acetic Acid, -phenylethyl Ester	C_10_H_12_O_2_	103-45-7	N.D	N.D	N.D	2.66 ± 0.07 a	N.D
H23	Ethyl Nonanoic	C_11_H_22_O_2_	123-29-5	N.D	N.D	N.D	1.51 ± 0.00 a	N.D
H24	Ethyl Acetate	C_4_H_8_O_2_	141-78-6	N.D	N.D	N.D	10.6 ± 0.16 a	N.D
H25	Succinic Acid, Cyclobutyl Ethyl Ester	C_10_H_16_O_4_	1,000,330-06-0	N.D	N.D	N.D	1.59 ± 0.01 a	N.D
H26	Oxalic Acid, Hexyl 2-phenylethyl Ester	C_16_H_22_O_4_	1,000,309-66-0	N.D	N.D	N.D	N.D	0.63 ± 0.02 a
H27	2,4-Di-tert-butylphenol	C_14_H_22_O	96-76-4	15.54 ± 0.55 a	0.70 ± 0.10 b	N.D	N.D	N.D
H28	Acetic Acid	C_2_H_4_O_2_	64-19-7	N.D	N.D	N.D	1.09 ± 0.08 b	24.27 ± 0.81 a
H29	Acetoin	C_4_H_8_O_2_	513-86-0	N.D	N.D	N.D	1.92 ± 0.08 a	N.D
H30	Heneicosane	C_21_H_44_	629-94-7	12.0 ± 2.18 a	N.D	N.D	N.D	N.D
H31	1-Chloropentane	C_5_H_11_Cl	543-59-9	N.D	57.00 ± 1.51 a	N.D	N.D	N.D
H32	Ammonium Acetate	C_2_H_7_NO_2_	631-61-8	N.D	N.D	2.01 ± 0.14 a	N.D	N.D
H33	Naphthalene, 1,2,3,5,6,8a-hexahydro-4,7-dimethyl-1-(1-methylethyl)-, (1S-cis)-	C_15_H_24_	483-76-1	N.D	N.D	0.44 ± 0.05 a	0.11 ± 0.00 b	N.D
H34	Bicyclo[5.2.0]nonane, 2-methylene-4,8,8-trimethyl-4-vinyl-	C_15_H_24_	242,794-76-9	N.D	N.D	1.16 ± 0.05 a	0.72 ± 0.02 b	0.74 ± 0.04 b
H35	Pentane, 1-(1-ethoxyethoxy)-	C_9_H_20_O_2_	13,442-89-2	N.D	N.D	N.D	0.33 ± 0.01 b	7.82 ± 0.46 a
H36	1,3,4-Oxadiazole	C_2_H_2_N_2_O	288-99-3	N.D	N.D	N.D	N.D	0.36 ± 0.04 a

N.D: not detected. a–d: Different letters in peer data indicate significant differences between groups (*p* < 0.05).

## Data Availability

The data used to support the findings of this study can be made available by the corresponding author upon request.

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
