# Peer review of "GC-MS Coupled with Rate-All-That-Apply (RATA) to Analyse the Volatile Flavor Substances of Yellow Wine during Fermentation"

_foods, 2023, doi:10.3390/foods12101992_

Round 1
Reviewer 1 Report
Abstract: Rewrite this section. First add 2 lines about the sample type, thereafter methodology used and results observed (numerical values). Add one recommendation and usefulness of the present study.
Introduction
Add a revised and dedicated section about the designing of present study. What insists the authors to perform the experimental work of present study?
Methodology
2.3 Consumer test: Why only 40 participants were involved in this test? The number should be more to validate the concept.
Table 2 C2H6O should be written as C2H6O
Check the formula's throughout the manuscript.
Antioxidant activity of the wine could be checked. This part is not covered by the Authors
Further, effect of storage time on sensorial attributes could be checked so that one can make recommendations accordingly.
Results and discussion
Add latest scientific findings related to the topic to improve the grace of discussion.
Author Response
We would like to thank you for your careful reading, helpful comments, and constructive suggestions, which has significantly improved the presentation of our manuscript.
We have carefully considered all comments from the reviewers and revised our manuscript accordingly. In response report, we summarize our responses to each comment from you.
We would like to thank the referee again for taking the time to review our manuscript.

Reviewer 2 Report
The manuscript entitled „GC-MS coupled with RATA analysis of volatile flavor substances during fermentation of yellow wine” describes a study on a sensory characterization of rice wine.
The study and the idea are interesting. But the manuscript requires a substantial amount of polishing.
Please explain RATA in the title – it is not a common abbreviation. Include in the title that it is a rice wine.
Remove “innovative” from the last sentence of the abstract. It is an exaggeration.
Line 38 – explain SPME
Line 48 – add abbreviation of SAFE
Line 61 – explain CATA
How old was wine at the last time point? It is not clear from the description.
Line 94: 65 m fibre? I think you have mistaken the unit.
The semi-calculation in response to a single IS is an incorrect approach for SPME. Please see the recent commentary which explains that (https://doi.org/10.1016/j.jfca.2022.104955). I am aware that it is a commonly used approach to report that in this way and there are hundreds of articles with such a calculation, but it is incorrect. Please remove it and stick to the peak areas which are more reliable than calculation in response to a single IS. Consequently, the ROAV calculation is also incorrect. Please remove it.
The same abbreviations are explained later on in the manuscript. Please order it somehow.
Have you received the ethical committee agreement for this sensory analysis?
Line 146: SPSS is not a method. It is software.
Don’t write about the identification of compounds if only NIST was applied. To talk about identification, all the compounds should be confirmed with authentic standards. Just based on NIST with a single Q MS it can be only tentative identification or annotation.
Line 169: what do you mean by amount here?
Line 171-173: “species” is not the correct word in this context.
Table 2. The numbers in formulas should be in subscripts.
Compounds are not equal to components! Don’t mix that.
Line 224: ?
Line 251: how is it possible that the rice product contains sulforaphane alcohol? It looks like an incorrect annotation of compounds. Sulforaphane derivatives can be obtained only in Brassica plants.
Line 307-308 – it is not clear where is the new compound. Please present it differently.
Line 307 - pentane,1-chloro-, phenylethyl alcohol – seems like an artefact, not a real VOC
Figure 7 – please improve the quality. Fonts cannot be read.
There is a lot of data presented but they are not discussed enough. Please add more scientific discussion of what you have obtained.
The majority of references are from the country of the authors. Please discuss your work more internationally. There are hundreds of fantastic studies from Europe on sensory analysis and volatilomics.
Some sentences have "missing parts" and the reader need to guess what the authors meant. Some words are used incorrectly. In generl, it would be better if the article would be edited by language expert.
Author Response
We would like to express our sincere gratitude to you for your comments. These comments are all valuable and helpful to revising and improving our manuscripts, as well as the important guiding significance to our study.
We have carefully considered all comments from the reviewers and have made correction which we hope meet with approval. In response report, we summarize our responses to each comment from the reviewers.
Please see response report and manuscript for specific details

Round 2
Reviewer 2 Report
In my opinion, the authors did a lot of work to improve the quality of the manuscript. All my comments have been carefully addressed.
My last minor comment, which can be changes at the proof level - please in the last sentence of section 2.2. replace for "each volatile compounds". I am allergic to volatile component.
I belive that the quality of manuscript improved.